# Preparation, Characterization and Magneto-Optical Properties of Sm-Doped Y_2_O_3_ Polycrystalline Material

**DOI:** 10.3390/mi13122254

**Published:** 2022-12-18

**Authors:** Andrzej Kruk, Krzysztof Ziewiec

**Affiliations:** Institute of Technology, Pedagogical University of Krakow, Ul. Podchorążych 2, 30-084 Kraków, Poland

**Keywords:** Y_2_O_3_, magneto-optical properties, transparent ceramics, arc plasma melting, luminescence

## Abstract

In this paper, physicochemical properties of pure Y_2_O_3_ and samarium (Sm)-doped Y_2_O_3_ transparent ceramics obtained via arc plasma melting are presented. Yttria powder with a selected molar fraction of Sm was first synthesized by a solid-state reaction method. High transparent yttria ceramics were obtained by arc plasma melting from both the pure and Sm oxide-doped powders. The morphological, chemical and physical properties were investigated by X-ray diffraction and scanning electron microscopy. The optical band gap was calculated from the absorption spectra so as to understand the electronic band structure of the studied materials. Samples indicate a series of luminescence bands in the visible region after excitation by laser light in the range from 210 to 250 nm. Magneto-optical measurements were carried out in the 300–800 nm range at room temperature. It can be seen that a maximum Verdet constant ca. 24.81 deg/T cm was observed for 405 nm and this value decreases with increasing wavelength. The potential usefulness of the polycrystalline material dedicated to optics devices is presented.

## 1. Introduction

Magneto-optical devices that use garnet crystals have been proposed for a wide range of applications, including for optical switches [1], optical communication systems [2], light modulators [3], quantum storage devices [4], nuclear magnetic resonance spectrometers [5], optical circulators [6], and magnetic-field sensors [7]. While they all show relatively high performance, they have not found much use in practice because of the difficulty in manufacturing the crystal and the high cost involved. However, it is possible to avoid these problems if doped yttria is used in the polycrystalline form in the devices cited above. Transparent yttria ceramics have been widely investigated because of their optical applications, including the visible (VIS) and infrared (IR) windows, and as host materials for solid-state lasers, fluorescent labels, multi-dimensional volumetric displays, and luminous pipes for high-intensity discharge lamps. Yttria (Y_2_O_3_) is a good candidate material to use in magneto-optical devices that retain a high chemical stability, large infrared cut-off, high corrosion resistance, absence of birefringence, and low absorbance coefficient in VIS–IR spectrum—while having a relative low cost [8,9]. Disadvantages of yttria sintered materials are their relatively weak mechanical properties, very high melting point (2700K) and UV absorption. The thermal conductivity of Y_2_O_3_ is 13.6 Wm/K at room temperature. Yttria crystallizes in a structure which corresponds to the C-type cubic type of the space group Ia3. Yttria can be C-bixbyite, cubic, hexagonal, or monoclinic crystal structure [10,11,12]. The unit cell of Y_2_O_3_ includes one site with point-group symmetry C3i and three sites with point-group symmetry C2. There is no center of inversion at the C2 site and the luminescence of incorporated lanthanide ions is predominantly associated with this site. Since C3i symmetry implies an inversion center, the electron-dipole transitions are forbidden. In this type of lattice, there are two distinct sites available for the substitution of trivalent ions. Hence, yttria has a high potential to create a solid state solution with rare-earth ions [12,13,14,15]. Several basic luminescence data of Sm^3+^-doped Y_2_O_3_ single crystals [16] and sinters [17] have been reported, but these studies do not cover polycrystalline ceramics obtained by arc plasma melting. Typically, the spectroscopy studies show that the Sm^3+^ ceramic samples display red emission from^4^G_5/2_→^6^H_7/2_ transition when excited by UV radiation [18].

Transparent yttria ceramics were fabricated from pure and RE-doped Y_2_O_3_ nanocrystalline powders. Nano-sized yttria powders were synthesized by various types of wet chemical methods e.g.: sol–gel [19], Peccini [20], EDTA [15], co-precipitation [21], combustion [22], hydrothermal [23], and solid state reactions [24]. The quality of the starting powder (nano-sized dimension, phase and chemical purity) and the doping have a strong influence on the optical performance of the yttria sinters obtained. Modern synthesis methods of yttria oxide materials—such as hot isostatic pressing [25], vacuum sintering [26], high-pressure/high-temperature [27], and spark plasma sintering (SPS) [28]—can be used to obtain high-transparency ceramics in both VIS and IR regions. Kruk [29] studied the optical performance of pure and Ce^3+^- or Pr^3+^-doped Y_2_O_3_ polycrystalline materials in the UV, VIS and NIR regions. It was found that the transmittance of the host material in the VIS region (380–760 nm) is higher than 51%, and translucence occurred in the UV region. Pure or synthetic MgO polycrystalline ceramics obtained by the arc-melting technique with 5 mm thickness achieved very good in-line transmission of over 74%, and of 60% in the VIS spectrum range [30,31]. Hu et al. prepared Dy_2_O_3_ transparent ceramics by vacuum sintering of nano-sized powders. The in-line transmittance values of the polycrystalline sample with 1.0 mm thickness are 75.3% at 2000 nm and 67.9% at 633 nm [32]. Yttrium aluminum garnets (YAGs) doped with Ce^3+^, Pr^3+^ and Cr^3+^ were prepared via a solid state reaction and sintered by the vacuum sintering technique. The highest transmittance of Ce^3+^, Pr^3+^ and Cr^3+^ doped YAG ceramic of 72% is reached with 0.4 mm thickness [33]. Transparent Y_2_O_3_ ceramics sintered by SPS show an in-line transmittance to 1 mm to be 80.6% at 1100 nm, which is close to the theoretical transmittance of 81.6% [34]. Generally, by increasing the Rare Earth (RE) doping elements in materials, the bandgap is decreased [35].

The optical behavior, which is indicated by a series of absorption bands in the VIS region of polycrystalline materials, depends strongly on the rare earth elements. In order to improve the transmittance in both VIS and IR wavelength ranges, the optimization of synthesis process parameters such us sintering temperature and holding time is sought. On the other hand, some sintering aids help the elimination of pores from ceramic materials, thus improving the transmittance of transparent ceramics. Pure yttria is proposed as an advanced Faraday material candidate due to its high optical properties and ability to incorporate the paramagnetic ions into a crystal lattice. Previous investigations of the doped Y_2_O_3_ indicate a high magneto-optical effect [10,36]. The Faraday rotation effect describes the rotation of a polarization plane of light (linear polarized) passing through a material located in an external magnetic field parallel to the direction of propagation of the light. In general, both a higher Verdet constant and lower sample thickness are necessary for device applications.

The main goal of this paper is to investigate the physicochemical properties of the Sm-Y_2_O_3_ polycrystalline ceramic including microstructural and optical properties. The absorbance coefficient and the Verdet constant of Sr-doped yttria polycrystalline ceramics were also measured for the evaluation of optical quality and magneto-optical behavior. 

## 2. Materials and Methods

### 2.1. Methodology 

Particle size distribution was measured using a Shimadzu SALD-7500nano using a static light scattering (SLS) technique with 405 nm laser. 

Imaging of surface morphology and chemical analysis were performed using a JEOL JSM-6610LV Scanning Electron Microscope equipped with an Energy Dispersive X-ray (EDS) detector manufactured by Oxford Instruments. To measure grain size, SEM images after binarization were processed using ImageJ 1.50 software.

The X-ray diffraction measurements were performed using X’Pert XRD (PANalytical) powder diffractometer, using CuKα radiation. The diffraction pattern was measured with a step of 0.08°, counting time of 10 secs/step and 2θ range 15°–85°. Rietveld quantitative phase analysis and phase identification were performed using High Score Plus v3.0e software by PANalytical. The crystallite size was estimated from the FWHM of the diffraction peaks using the well-known Scherrer’s equation [37]:(1)$$D_{XRD}=\frac{K\cdot \lambda }{\beta \cdot cos\theta }$$
where *K* is the shape coefficient for the reciprocal lattice point (*K* = 0.89), *λ* is the wavelength of the X-rays (CuKα = 0.1788965 nm), *θ* is the peak position and *β* is the is full width at half maximum of the diffraction peak by instrumental broad factor.

The average crystallite size was calculated from nine major XRD peaks. 

A minimum dislocation density of the particular structure was estimated using the relation from [38]:(2)$$\rho \approx \frac{1}{D_{xrd}{}^{2}}$$
where *D_xrd_* is the calculated crystallite size. 

To measure the Verdet constant, a magneto-optical spectrometer manufactured by the Instytut Fotonowy (Kraków, Polska) was used. This measurement setup uses a stable magnetic field in the range from 0 to 0.02 T generated by a Helmholtz coils system. A balanced polarimeter is used to determine the angle through which plane-polarized light has been rotated by a polycrystalline material. This unit is basically composed of the Wollaston prism and two identical photodiodes. The photodiode signals are digitized and sent to Lock-in amplifier before being collected on PC for further analysis. The device was used to measure Verdet constants as a function of wavelength (400–800 nm with 50 nm step). Samples were isolated from any external magnetic field using a Faraday cage. Measurement error was 10%. 

The reflected luminescence spectra were obtained using a pulsed OPO (optical parametric oscillator) laser system made by EKSPLA (model PT304, Vilnius, Lithuania). Indirect excitation of samples was performed between 210 and 400 nm. The emission signal was recorded by the SilverNova Stellarnet spectrometer (Stellarnet, Tampa, FL, USA) with a spectral resolution of 0.5 nm; or by high resolution Oxford Mechelle 5000 spectrograph coupled with Andor 300 series CCD camera (Yatton, Bristol, UK). The 405 nm continuous wave (cw) diode laser (Spectra-Laser, Opole, Poland) was used as an optical pumping source in the luminescence study. The luminescence signal and chromaticity were detected by the integrated sphere IC2 (Stellarnet) coupled to the SilverNova StellarNet spectrometer. 

The absorption spectra were recorded with a set-up consisting of a deuterium-halogen lamp (Spectra-Laser) as light source and the SilverNova StellarNet spectrometer connected to a computer. All measurements were conducted at room temperature.

### 2.2. Sample Preparation

High purity commercially available micro sized powders of Y_2_O_3_ (Merck, 99.999%) and Sm_2_O_3_ (Merck, 99.998%) were used as starting raw materials. According to the stoichiometric formula of Sm_0.01_Y_1.99_O_3_, the weighted powders were homogenized in absolute ethyl alcohol for 10 minutes with zirconia balls. After mixing, the slurries were dried. The homogeneous powders were then dry-pressed at 13 MPa into disks 15 mm in diameter and 5 mm in thickness. A two-step sintering process of the green bodies was conducted in ambient air. First, the samples were heated at a rate of 6 °C/min at a relatively low temperature (T = 1000 °C). After cooling to room temperature at a rate of 15 °C/min, the samples were further melted. Next, the obtained Sm:Y_2_O_3_ ceramics were melted at ca. 2760 °C for 10 minutes. No additional sintering to remove oxygen vacancies was necessary. At the end, the ceramics were mirror polished on both surfaces. Transparent, dense, Sm_0.01_Y_1.99_O_3_ polycrystalline samples were successfully sintered.

## 3. Results and Discussion 

### 3.1. Powder Characteristics

Figure 1 shows the particle-size distribution of Sm_0.01_Y_1.99_O_3_ powder after blending. The investigated powder exhibits unimodal size distribution with medium diameter (D_50_) of 20 μm. The powder particle size was in the wide range of 0.1–80 μm. However, there was agglomeration of some of the particles. 

### 3.2. Bulk Sample Characteristics

Figure 2 exhibits the XRD patterns of the Y_2_O_3_ and Sm_0.01_Y_1.99_O_3_ bulk samples obtained after melting. Both polycrystalline materials possess the perovskite structure without any detectable secondary phase in the XRD patterns, which demonstrates a homogeneous solid solution. The XRD patterns corresponding to yttria-stabilized zirconia were not detected even after adding the ball milled of powder to zirconia balls. The calculated values of parameters including lattice spacing d_hkl_, lattice parameter a, unit cell volume V and calculated crystallite size D_xrd_ and dislocation density ρ of the pure and doped sample are presented together in Table 1. 

The obtained material has a cubic phase structure with the Ia-3 space group according to the ICDD XRD PDF Card No. 98-008-1861. Magnified diffraction peaks near 50° are displayed as an inset in the Figure 2. The shifting of peaks can be distinctly observed between two samples. Line broadening and shifting along with 2θ dependence of FWHM indicate the Sm ion contributions. The average crystallite size of Sm-doped Y_2_O_3_ obtained by arc melting technique is higher than that of pure Y_2_O_3_ (Table 1). Lattice parameter differences were found in the XRD spectra between those of pure and of Sm-doped yttria, confirming that the incorporation of 1 mol% Sm did affect the structure of the host. The dissolution of the dopant whose ionic radius (1.02) is larger than that of the host ions (0.96) creates stresses in a sample, and these conditions lead to local structural instability [39]. The measured lattice parameters of pure yttria are consistent with the theoretical reference value for Y_2_O_3_ of 10.604 Å.

Figure 3 presents the morphology of the surface and cross section of the Sm_0.01_Y_1.99_O_3_ sample obtained after the arc plasma melting procedure. The investigated sample was cracked to reveal the microstructure in cross section. The SEM image shows the dense structure of the ceramics at 1000× magnification. The morphology of the surface revealed a coarse-crystalline and compact grain structure. The grain boundaries were only weakly visible; however, the major radius of sharp-edged grains were from 0.1 to 35 µm. The crystallite size obtained from XRD analysis is different to the particle size determined from SEM images, because a particle may be made up of several different crystallites, or be just one single crystallite. Crystallites are formed as the result of fast cooling of fully melted ceramic material. As shown in Figure 3, the typical microstructure of an arc-melted specimen can be described as piles of stacked sheets parallel to the plane defined by the crystallographic axis. A very small number of crystallites displayed separated pores with a diameter that did not exceed 1 nm. When a VIS–NIR light beam was transmitted through the samples, the Sm:Y_2_O_3_ polycrystalline ceramic exhibited high transparency, which could be ascribed to the grain boundaries without any pores inside. Free electrons are trapped in localized artifacts, at the grain boundary, in impurities and defect points, and on irregular surfaces. Free electrons are able to interact with visible photons that results in the relatively low transmittance of the material.

We identified artifacts from the SEM images using ImageJ software based on the morphology of the material, and then conducted the statistical analysis using the ImageJ software. The obtained densities were above 99.8% and 99.9% for the Y_2_O_3_ and Sm_0.01_Y_1.99_O_3_ specimens, respectively. Results of EDX microanalysis for this composition are presented in Figure 4. EDX images present the spatial distribution of different elements in the sample. The EDX elemental mapping of the polycrystalline material confirmed the uniform concentration of the Sm, Y and O ions on the surface and throughout the cross-section. The X-ray microanalysis confirmed the presence of samarium, yttrium and oxygen, in the assumed proportion. The percentage weight ratio of Sm to Y ions is found approximately to be 0.6:75.6 and these results showed a partial agreement with the assumed chemical composition expressed as Sm_0.01_Y_1.99_O_3_.

Figure 5a shows the absorption coefficient and absorption cross-section spectrum of the Sm_0.01_Y_1.99_O_3_ sample in the 200 to 1100 nm wavelength range. The optical and magneto-optical properties of the Y_2_O_3_ arc plasma melted sample were reported elsewhere [29,40]. The in-line transmittance is higher than 65% for the Sm_0.01_Y_1.99_O_3_ sample (Figure 5b) though decreases slightly in the visible region, mainly attributed to surface roughness which results in reflecting, scattering, and refracting of the photons [40]. Unfortunately, the photography was taken after having broken the as-obtained sample to reveal its internal details (SEM observations). The surface cracking patterns decrease the transmittance of the sample.

Due to Mie scattering theory, a wavelength increase leads to an increase in transparence [41]. Compared with the shift by the pure Y_2_O_3_ ceramic sinter of 230 nm reported by Zhang [42], the UV absorption edges of the sample doped with Sm^3+^ ion shifted 260 nm. The UV absorption edges of pure Y_2_O_3_ polycrystalline material reported elsewhere were of 350 nm [39]. The pure Y_2_O_3_ crystal exhibits both a stable crystal field and constant band-gap according to the energy band theory. The use of two steps synthesis and the improved procedure of arc plasma melting provide a new optimized strategy for the fabrication of high transparent ceramic with suitable optical properties. Replacing the powder with compact sinters reduces material wastage in contact with the plasma. Nevertheless, the Sm doping into the yttria matrix disturbs the crystal field, and induces impurity energy levels into the forbidden band, resulting in red-shift of the absorption edge. The Lambert Beer’s laws were used to compute the absorption coefficient *α* and absorption cross-section *σ* of Sm_0.01_Y_1.99_O_3_:(3)$$\alpha =\frac{-2.303\mathrm{lg}\left(\frac{I}{I_{0}}\right)}{L}$$
(4)$$\sigma =\frac{-2.303\mathrm{lg}\left(\frac{I}{I_{0}}\right)}{L\cdot N}$$
where *α* is the absorption coefficient, lg(*I*/*I*_0_) is the experimentally obtained optical density, *L* is the light path in the transparent sample and *N* is the number of doping (active) ions in the unit volume. The value of active ions *N* is 1.34·10^20^ ions/cm^3^ for the Sm_0.01_Y_1.99_O_3_ transparent sample.

The absorption coefficient spectrum of the Sm_0.01_Y_1.99_O_3_ consists of several overlapped broadened absorption bands placed mostly in the UV region. The observed absorption bands correspond to transitions from the ^6^H_5/2_ ground state of a molecule to the various excited levels as shown in Figure 5a. Detailed energy level assignments have been given for Sm ions by Carnall et al. [43]. The small peak observed at ca. 660 nm is related to the Fulcher band emission from a deuterium lamp (see Section 2.1).

The information about the electronic structure in the optical band gap of the polycrystalline materials can be obtained from the shape of the absorption edge in the Tauc plot. Figure 6 shows the Tauc plot for the Sm:Y_2_O_3_ transparent polycrystalline material calculated from the spectrum presented in Figure 5a. An absorption coefficient (*α*) exhibits exponential dependency on the photon energy and can be expressed by the Tauc plot rule as presented in [44]:(5)$${\left(\alpha \cdot h\nu \right)}^{1/\gamma }=B\left(h\nu -E_{g}\right)$$
where *h* is the Planck constant, *ν* is the photon’s frequency, *E_g_* is the band gap energy, and *B* is a constant. The *γ* factor depends on the character of the electron transition and is equal to 2 for the indirect transition band gaps.

The optical bandgap Eg computed from the experimental absorbance spectra was estimated to be 4.62 eV, with good agreement with a previously reported value for RE-doped Y_2_O_3_ [45]. The experimentally-found Eg values for pure and RE-doped Y_2_O_3_ estimated by optical absorption methods vary significantly, from 2.39 to 5.9 eV [39,45,46,47]. Yttria transparent ceramic shows deviation in the optical band gap due to the presence of Sm^3+^ ions in the host. 

According to the Dimitrov–Sakka equation, it is possible to determine the refractive index *n* from the optical band gap energy, as given by [48]:(6)$$\frac{\left(n^{2}-1\right)}{\left(n^{2}+2\right)}=1-\sqrt{\frac{E_{g}}{20}}$$

The refractive index was determined as 2.06. This computed value is similar to that of pure Y_2_O_3_ reported by Nigara [49]. 

Figure 7a shows an overview of emission spectra of the Sm_0.01_Y_1.99_O_3_ recorded in the wavelength region 500–800 nm excited by the selected wavelength from 210 to 250 nm. The obtained spectra exhibit three prominent broad bands located in the VIS light region which originated from different excitation transitions of the Sm ions.

Figure 7b shows the series emission peaks with maxima at 566 nm, 609 nm, and 669 nm, which are correlated to the ^4^G_5/2_→^6^H_J/2_ (J = 5, 7, and 9) transitions of Sm^3+^, respectively. All observed terms originate from the f-f transition of the 4f electron in Sm ions. The sample excited with deep UV 228 nm (5.51 eV) light showed the highest luminescence intensity. With an increase in excited energy, the observed intensity of luminescence grows, then decreases because the excitation energy is too high and leads to an increase in non-radiative energy transfer through the cross-relaxation channel, this being the distance between Sm^3+^ ions [43,50].

Figure 7c shows the emission spectrum from the Sm_0.01_Y_1.99_O_3_ sample collected on the 2-D Andor CCD operated with the Mechelle spectrograph A broad circle depicted the visible luminescence band of Sm^3+^ ions is observed under cw UV excitation in the sample. A notch UV filter was used to remove a pump light from the fluorescence signal. The key observation is that the luminescence beams had a circular and Gaussian shape with a diameter of 50 pixels. The photoluminescence measurements remained in agreement with the spectrum shown in Figure 8.

The recorded fluorescence spectrum (collected with the help of the integrated sphere method) under cw blue 405 nm light excitation is presented in Figure 8. From the obtained photoluminescence spectrum, the emission from the Sm_0.01_Y_1.99_O_3_ polycrystalline material is in green (575 nm), orange (610 nm) and red (660 nm) region. 

The quantum efficiency of the Sm_0.01_Y_1.99_O_3_ polycrystalline was determined using the following expression:(7)$$\eta_{QE}=\frac{\int^{\;}L_{S}}{\int^{\;}E_{R}-\int^{\;}E_{S}}*100\%$$
where *L_S_* it the total area of the emission spectrum, and *E_R_* and *E_S_* are the total area under excitation spectra with and without the sample, respectively. The experimentally-determined value of quantum efficiency for the arc plasma melted specimen is 48.43%.

The CIE 1931 color space chromaticity diagram is shown in Figure 9. The CIE coordinates of the investigated material was obtained by the StellarNet Software. The coordinates were calculated as X = 0.318, Y = 0.34 after excited white light (a deuterium + halogen lamp) and X = 0.492, Y = 0.362 under cw 405 nm excitation wavelength, which fall exactly in the red region as shown in the diagram. The inset Figure 8 shows the photography of the surfaces of the samples during radiation exposure by a cw 405 laser, photographed through a filter (450 nm).

To express the rotation of the plane of polarization of the light beam (θ) as a function of the wavelength, the following equation is used:θ = V·B·L(8)
where V is the Verdet constant, L is the optical path, and B is the magnetic induction component parallel to the light beam.

The Verdet constant is a function of the angular frequency of the incident light. Bacquerel has proposed the relationship between the Verdet constant and diamagnetic properties of the materials:(9)$$V=-\frac{e\lambda }{2mc^{2}}\frac{dn}{d\lambda }$$
where *c* is the speed of light, *e* and *m* are the charge and mass of the free electron, respectively, *dn*/*dλ* is the dispersion, and *V* is the Verdet constant. 

The useful parameter is the magneto-optical figure of merit (FOM) (Figure 10), as defined by the following Equations [50,51,52,53]:(10)$$FOM=\frac{\left|V\right|}{\alpha }$$
where *V* is the Verdet constant (deg/T∙m) and α the absorption coefficient (cm^−1^).

Results of the Verdet constant dispersion of Sm_0.01_Y_1.99_O_3_, performed in the range 400–800 nm, are shown in Figure 10. The Verdet constant continually decreases with wavelength. On the other hand, the Verdet constant is able to increase with increasing dispersion of the medium—Equation (9). Generally, the Verdet constant of the sample is lower than that of the typical commercial TGG single crystal in the VIS spectrum range. Paramagnetic rare earth ions such as Tb, Dy, and Pr can greatly increase the Verdet constant by doping into/becoming constituents of the Y_2_O_3_ lattice, which shorten the required sample size and reduce thermal effects more effectively [54]. In magneto-optical crystals, the Verdet constant of terbium gallium garnet (TGG) reaches up to −134 rad/(T⋅m) at the wavelength of incident light at 633 nm owing to its high concentration of highly paramagnetic ally Tb^3+^ ions [55]. In addition, TGG has a high transmittance and good thermo-optical characteristics enabling operation at high average power [55,56]. Nevertheless, crystal preparation is costly, time-consuming, difficult, and linear birefringence can easily occur. Alternatives are various series of glasses, including, tellurate [57], heavy metal oxide glass [58], chalcogenide glass [59], and phosphate [60]. The transparent TGG, yttria alumina garnet (YAG) and terbium alumina gallium garnet (TSAG) crystals or ceramics, after being doped with RE ions such as Ce^3+^, Pr^3+^, Nd^3+,^ Dy^3+^, Ho^3+^, and Tm^3+^ have been found to be able to effectively improve the Verdet constant [61,62,63,64]. The transparent magneto-optical (Tb_1−x_Pr_x_)_3_Al_5_O_12_ ceramics—obtained by a co-precipitation method and sintering—demonstrated that Pr^3+^ doping would lead to a ~10 % larger Verdet constant than independent TAG [8]. Kruk fabricated a transparent Ce^3+^ or Pr^3+^ doped Y_2_O_3_ polycrystalline material and their Verdet constant is ~2 times that of pure Y_2_O_3_ [29]. The Verdet constants of the selected materials are presented in Table 2. It should be emphasized that Tb^3+^ ions exhibit the highest magnetic moment and paramagnetic effects because of their electronic transition of 4fn →4fn-15d. However, the results of Sm:Y_2_O_3_ indicate that polycrystalline materials have potential application in optical laser systems. The arc plasma melting procedure allows the large size sample dedicated to high power laser systems to be obtained.

The MO FOM parameter of the investigated sample was presented inset in Figure 10. Despite the impact of the absorbance coefficient, the highest values of FOM correlated with the Verdet constant. The absorbance bands are placed behind the selected wavelength.

Based on the values presented in Table 2, there is a significant difference between the samples doped with Tb, and those doped with other paramagnetic ions.

## 4. Conclusions

The present paper reports the synthesis and characterization of the Sm-doped Y_2_O_3_ transparent ceramic obtained by arc plasma melting technique. This powder was prepared by solid state reaction method. The prepared transparent Y_2_O_3_-based ceramics show good optical transparency, homogeneity, chemical stability and high density. The in-line optical transmittance of Sm:Y_2_O_3_ at a wavelength of 500 nm achieves a transmittance of 51.2% with 3 nm thickness. XRD results reveal that the foreign metal cations are structurally incorporated into the crystal lattice of Y_2_O_3_. In the UV–VIS spectrum range, the Sm_0.01_Y_1.99_O_3_ material shows absorption bands, which is important in the design of magneto-optical devices at a wavelength close to this absorption band. The photoluminescence emission intensity was at a maximum around 610 nm that results in ^4^G_5/2_ -_6_H_7/2_ transition within Sm; and emits red light at 610 nm under deep-UV light excitation. This polycrystalline material might be used for color tuning in many applications due to the high QE parameter. Sm ions improved the magneto-optic effect in the investigated material. Further studies are necessary to continue steadily to increase the Verdet constant and decrease the optical absorbance coefficient. However, the arc melted specimen can be shaped more freely and their productivity is more efficient than a single crystal.

## Figures and Tables

**Figure 1 micromachines-13-02254-f001:**
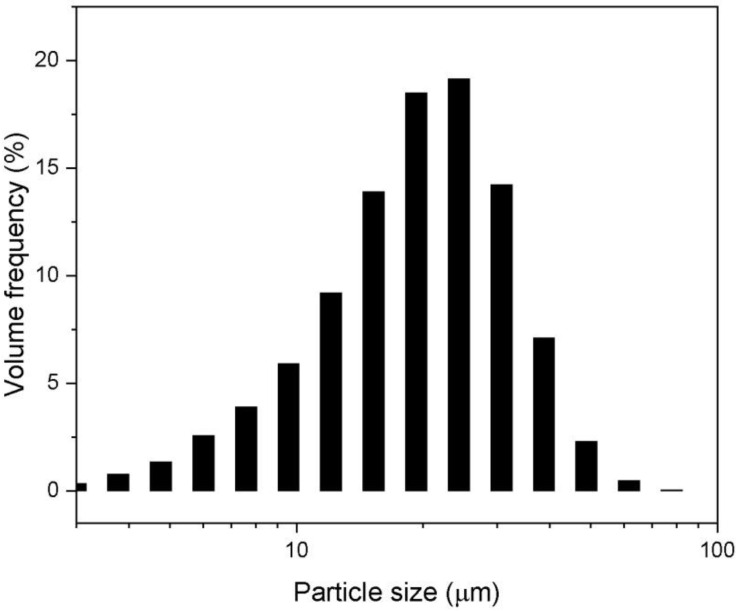
The particle-size distribution of Sm_0.01_Y_1.99_O_3_ powder after blending.

**Figure 2 micromachines-13-02254-f002:**
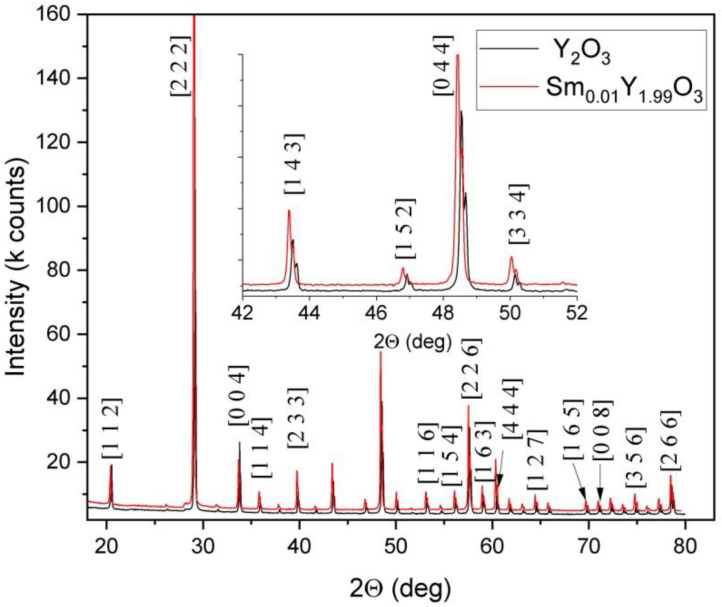
The XRD diffraction peaks of Sm_0.01_Y_1.99_O_3_ powder after blending.

**Figure 3 micromachines-13-02254-f003:**
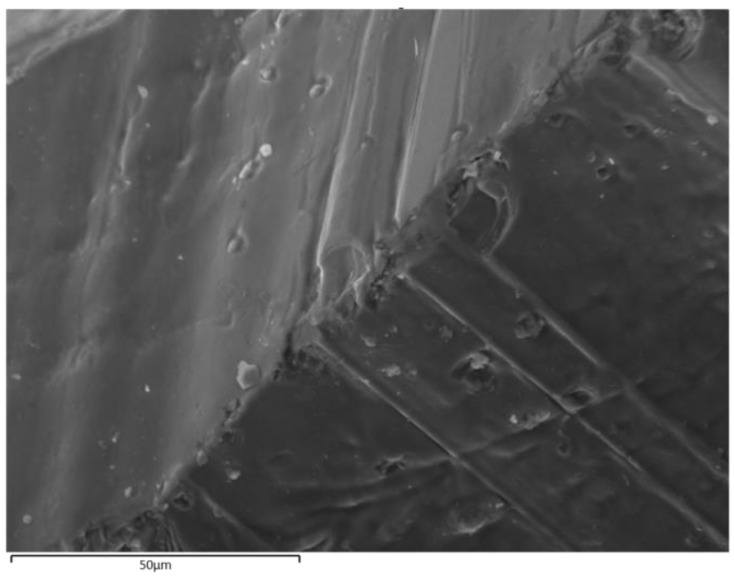
The SEM morphology of the cross section of the Sm_0.01_Y_1.99_O_3_ sample obtained after the arc plasma melting procedure.

**Figure 4 micromachines-13-02254-f004:**
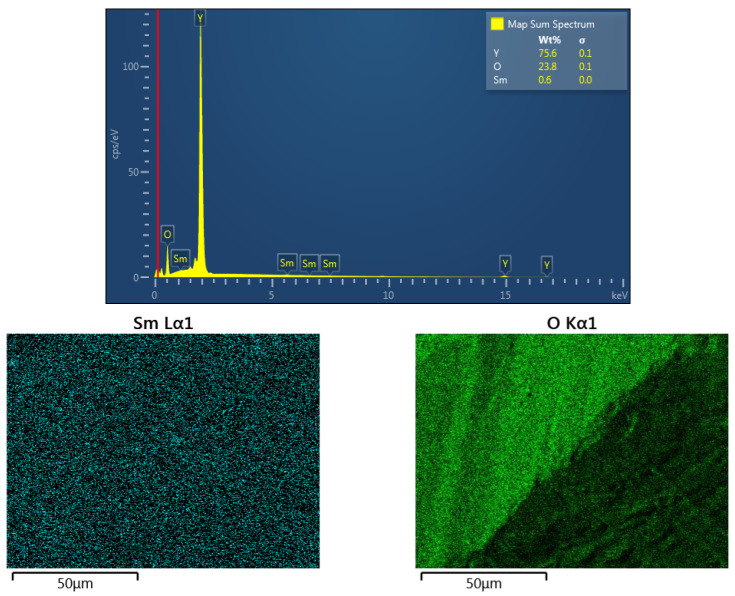
The EDS spectrum and elemental distribution maps of the Sm_0.01_Y_1.99_O_3_ sample obtained after the arc plasma melting procedure.

**Figure 5 micromachines-13-02254-f005:**
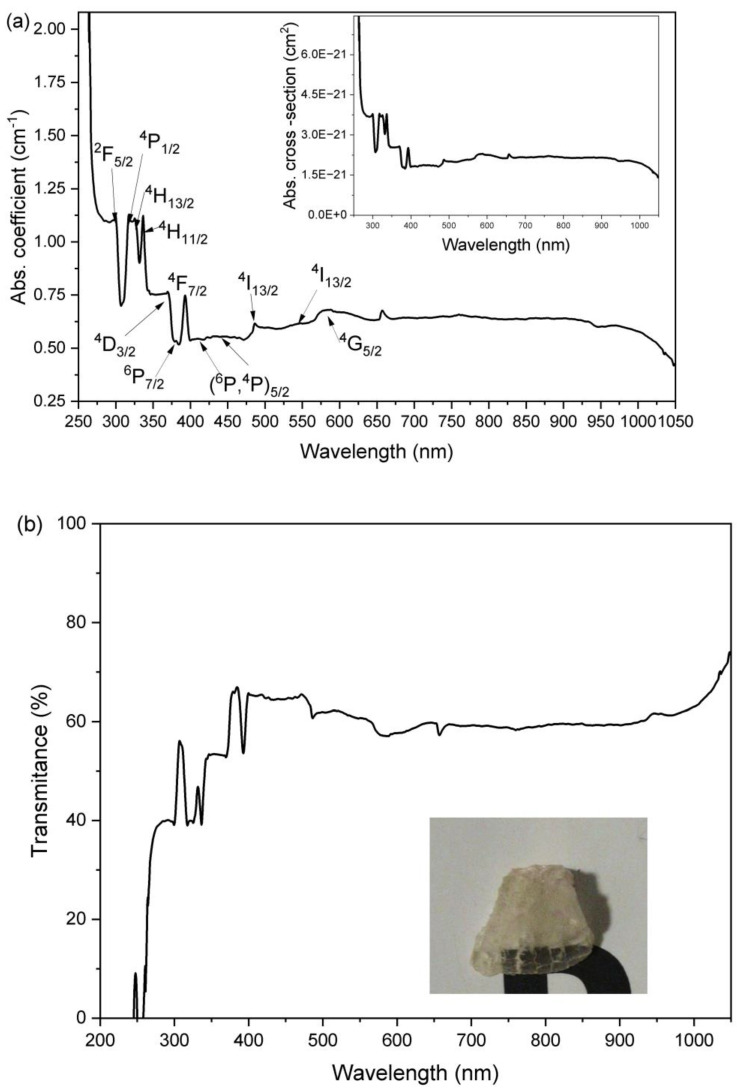
(**a**) The room-temperature absorption coefficient spectrum in the UV–VIS–NIR region for the Sm:Y_2_O_3_ transparent ceramics. Inset: the absorption cross section coefficient in the UV–VIS–NIR region. (**b**) The in-line transmittance spectrum in the UV–VIS–NIR region. Inset: the photo of the Sm_0.01_Y_1.99_O_3_ sample after being broken for the SEM observations.

**Figure 6 micromachines-13-02254-f006:**
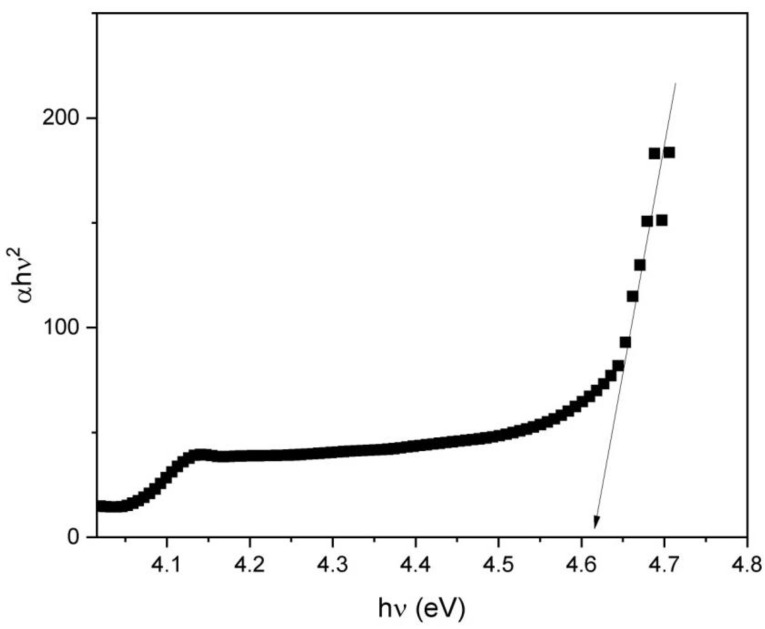
Tauc plot ((αhν)^2^ vs. energy) for the Sm:Y_2_O_3_ transparent polycrystalline.

**Figure 7 micromachines-13-02254-f007:**
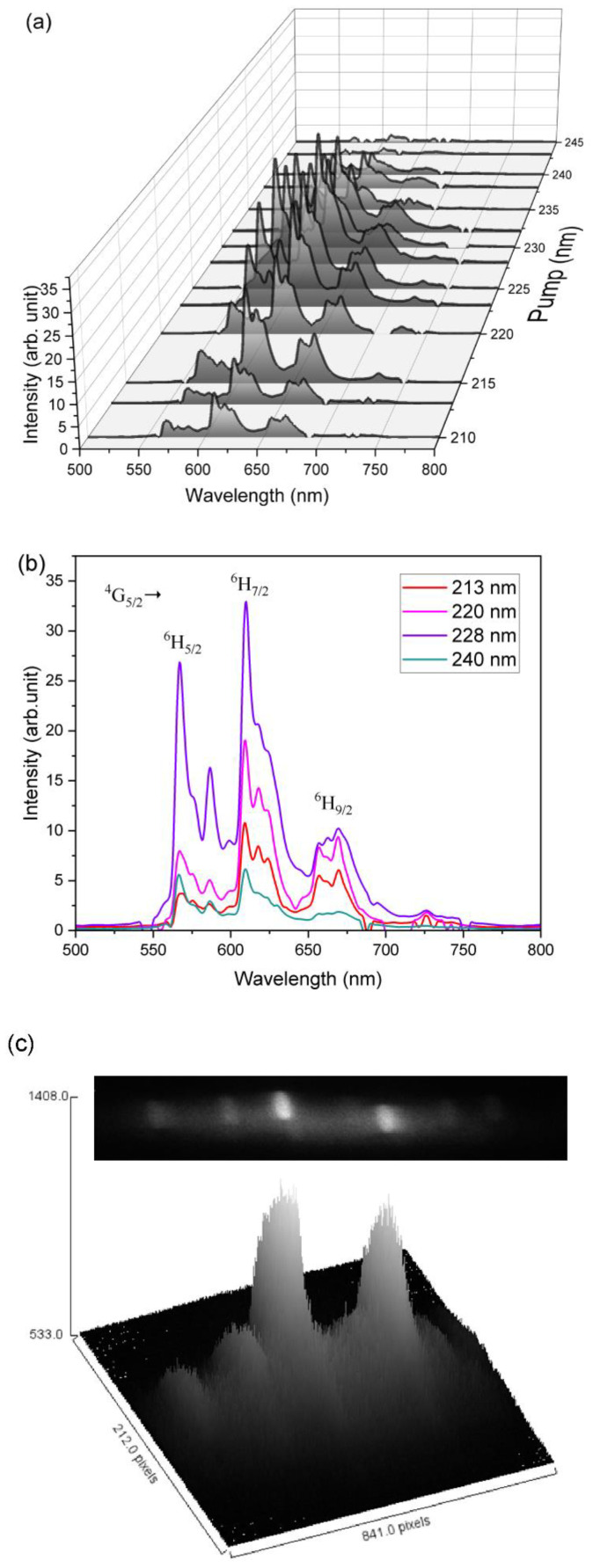
(**a**) The luminescence spectra of the Sm_0.01_Y_1.99_O_3_ excited by the selected emission wavelength from 210 to 250 nm. (**b**) The luminescence spectra of the Sm_0.01_Y_1.99_O_3_ excited by the four different wavelengths. (**c**) Recorded brightness image of luminescence emission from a CCD camera and its corresponding emissivity profile.

**Figure 8 micromachines-13-02254-f008:**
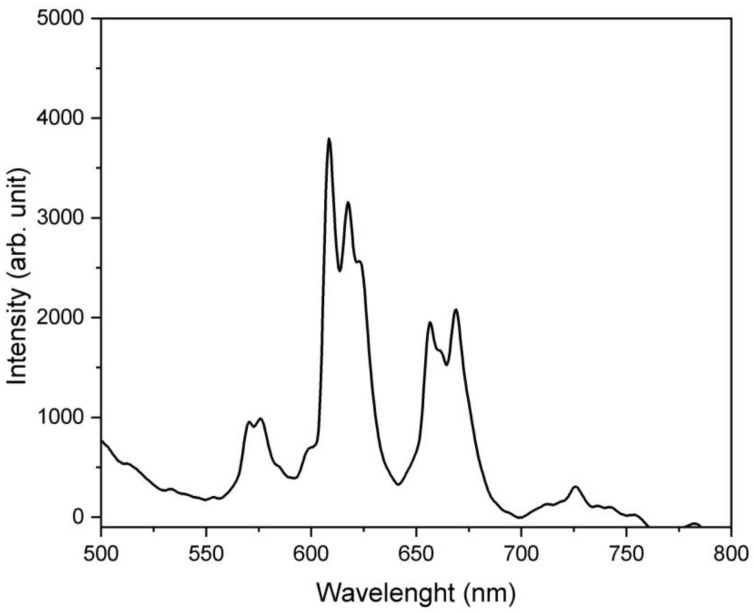
The luminescence spectrum of the Sm_0.01_Y_1.99_O_3_ excited with 405 nm cw excitation wavelength.

**Figure 9 micromachines-13-02254-f009:**
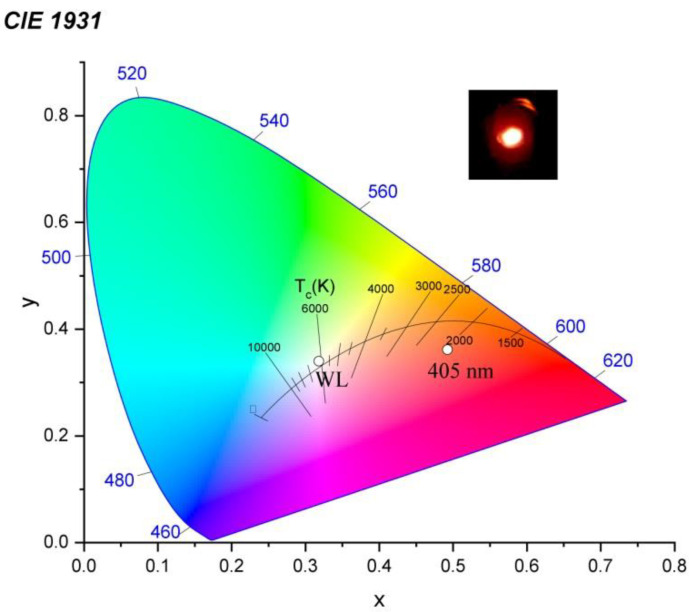
CIE color coordinates 1931 of Sm_0.01_Y_1.99_O_3_ illuminated by white light (“WL” dot) and 405 nm excitation (“405 nm” dot).

**Figure 10 micromachines-13-02254-f010:**
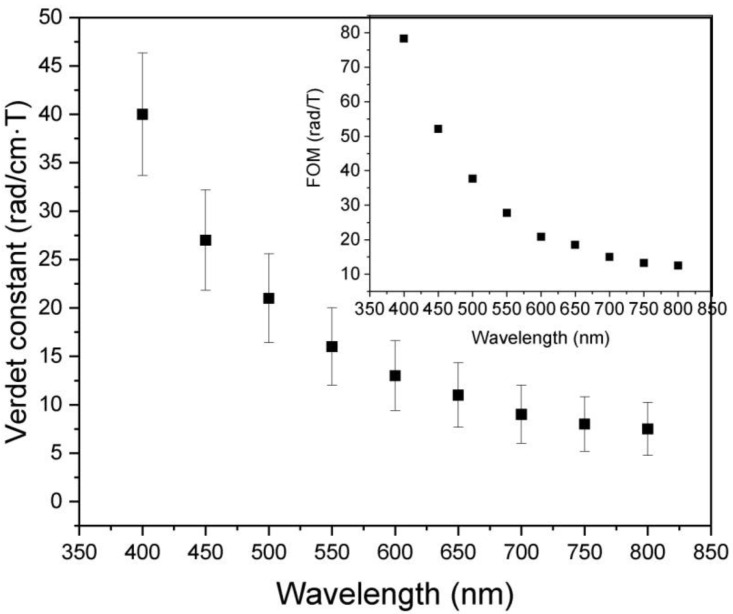
Wavelength dependence of Verdet constant of Sm_0.01_Y_1.99_O_3_.

**Table 1 micromachines-13-02254-t001:** The value of the lattice spacing d_hkl_, the lattice parameter a, unit cell volume V with crystallite size D_xrd_ and dislocation density ρ of the Y_2_O_3_ and Sm_0.01_Y_1.99_O_3_ bulk sample.

Y_2_O_3_					Sm_0.01_Y_1.99_O_3_				
d_222_ [nm]	A [Å]	V [Å^3^]	D_xrd_ [nm]	ρ [10^14^ m^−2^]	d_222_ [nm]	A [Å]	V [Å^3^]	D_xrd_ [nm]	ρ [10^14^ m^2^]
27.01	10.6	1191.02	15.05	4.14	31.2	10.62	1197.77	28.62	12.2

**Table 2 micromachines-13-02254-t002:** Verdet constants of various MO materials.

Sample	State	Verdet Constant			Ref
Ho_2_O_3_	ceramic	−180 rad/Tm (632.8 nm)	−46 rad/Tm (1064nm)	20 rad/Tm (1550 nm)	[63]
(Ho_0.9_Pr_0.1_)_2_O_3_	ceramic	−235 rad/Tm (635 nm)	−82 rad/Tm (1064 nm)	-36 rad/Tm (1550 nm)	[53]
(Dy_0.9_Y_0.05_La_0.05_)_2_O_3_	ceramic	886 rad/Tm (405 nm)	444 rad/Tm (532 nm)	101 rad/Tm (1064 nm)	[64]
Tb_2_O_3_	ceramic	~480 rad/Tm (632nm)	~130 rad/Tm (1075 nm)	~40 rad/Tm (1561nm)	[36]
30%Tb_2_O_3_:Y_2_O_3_	ceramic	~220 rad/Tm (632nm)	~42 rad/Tm (1075 nm)	~17 rad/Tm (1561nm)	[36]
Tb_2_Hf_2_O_7_	ceramic	~270 rad/Tm (532nm)	~150 rad/Tm (650 nm)	50 rad/Tm (1064 nm)	[65]
T40	glass	−158 rad/(T⋅m) (633 nm)	− 48.87 rad/Tm (1064 nm)		[56]
15%TiO2	glass	~0.225 min/G·cm (532 nm)	~0.173 min/G·cm (632 nm)	~0.168 min/G·cm (650 nm)	[57]
TGG	monocrystalline	~1.2 min/G·cm (460 nm)	~0.75 min/G·cm (532 nm)	~0.375 min/G·cm (695 nm)	[66]
TSAG	monocrystalline	~2 min/Oe·cm (405 nm)	~1 min/Oe·cm (500 nm)	~0.2 min/Oe·cm (500 nm)	[67]

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
