# Peer review of "Preparation, Characterization and Magneto-Optical Properties of Sm-Doped Y2O3 Polycrystalline Material"

_micromachines, 2022, doi:10.3390/mi13122254_

Round 1

Reviewer 1 Report

The manuscript describes the fabrication and characterization of transparent Y2O3 and Sm–doped Y2O3 ceramics obtained via an arc plasma melting route. The conception contains some interesting results, but there are some problems within this manuscript. Therefore, I could support its publishment after the authors address these questions as follows:

1. The authors should provide some information on polycrystalline transparent ceramics, including appearances and in-line transmittances.

2. Please check the validity of your cited references. For example, regarding the calculation on minimum dislocation density, I did not find the formula (Eq. 1) in your ref. 28. In addition, please also check your calculated results for minimum dislocation density.

3. The crystal indices in Fig. 2 should be marked.

4. “The optical bandgap Eg computed from the experimental absorbance spectrum was estimated to be 4.62 eV, in good agreement with previously reported value for pure Y2O3 [33]”. However, the optical band gap determined by ref. 33 is 2.39 eV. It is not similar. What is the reason?

5. Please introduce a table comparing the Verdet constant values of your materials with related materials, such as Tb3Ga5O12, Ho3Al5O12, Ho2O3, Dy2O3, etc.

6. Figure of merit (FoM) for Verdet constant is generally used to assess the magneto-optical material properties. Thus, FoM should be calculated and discussed. Some references may be helpful for your calculation and discussion (DOI:10.1016/j.ceramint.2019.04.203; DOI: 10.1016/j.jeurceramsoc.2021.04.016).

7. There are some grammar mistakes and the English expression throughout the whole manuscript should be further improved.

  Above all, I would recommend its publishment after minor revision.

Reviewer 2 Report

In this manuscript entitled "Preparation, characterisation and magneto-optical properties of Sm doped Y2O3 polycrystalline material", the authors studied the structure, morphology, chemical and physical properties by XRD, FT-IR spectroscopy and SEM measurements. The maximum Verdet constant ca. 24.81 deg/T cm was observed for 405 nm and this value decreases with increasing wavelength. The research content is rich, but the studies and discussion of structure are still lacking. Therefore, this paper should be reconsidered after some revisions.

1. Miller indices of XRD peaks should be marked in the XRD.

 2. A recently updated article [Hui Zhang, Yan Wang, Haiou Wang, Dexuan Huo, and Weishi Tan, Journal of Applied Physics 131, 043901 (2022)] related to structure, morphology (particle size distribution and elemental distribution maps), and optical bandgap Eg of rare-earth oxide materials is suggested to be cited in the introduction or discussion part.

3. The authors claim that “Dxrd is the calculated crystallite size.” So how do you get the value of Dxrd? In addition, why is the crystallite size nanometer, while the particle size obtained by SEM is micrometer?

 4. Lattice spacing dhkl. what is the hkl? For example, d001? d 110?

 5. The two words “EDS” and “EDX” are used in many places in the article, which one should be used?

Round 2

Reviewer 1 Report

1. My original question (1) is that you should provide some information on your transparent ceramic samples, including the appearances and the in-line transmittances. It does not mean to review recent reports. Because your preparation method may be a good way for yielding highly transparent ceramics, please show your products in your manuscript.

2. Please check the validity of your cited references. For example, regarding the calculation on minimum dislocation density, I did not find the formula (Eq. 2) in your ref. 39. In addition, please also check your calculated results for minimum dislocation density, including units.

3. Some problems in upper and lower case should be corrected. I recommend its publishing after minor revision.

Author Response

  1. My original question (1) is that you should provide some information on your transparent ceramic samples, including the appearances and the in-line transmittances. It does not mean to review recent reports. Because your preparation method may be a good way for yielding highly transparent ceramics, please show your products in your manuscript.

Answer: The in-line transmittance and the photography of the sample were added to the manuscript (Fig.5b) . 

Also the text was added to the manuscript:

Unfortunately, the photography was taken after broken the as-obtained sample to reveal its internal details (SEM observations). The surface cracking patterns decrease the transmittance. 

2. Please check the validity of your cited references. For example, regarding the calculation on minimum dislocation density, I did not find the formula (Eq. 2) in your ref. 39. In addition, please also check your calculated results for minimum dislocation density, including units.

Answer: We checked all the literature again. Now in the reference [39] there is the corresponding equation.

[39]  A. Youvanidha, B. Vidhya, P. I. Nelson, Investigation on the structural, optical and electrical properties of ZnO- Y2O3 (YZO) thin films prepared by PLD for TCO layer applications, AIP Conference Proceedings, 2166, (2019) 

  • 10.1063/1.5131610

page 020023-3

3. Some problems in upper and lower case should be corrected. I recommend its publishing after minor revision.

Answer: We improved the manuscript including reference part. 

Thank You again for your valuable questions and suggestions. 

Reviewer 2 Report

The paper has been improved and can be accepted in current form.

Author Response

Thank you for your valuable review